# Raman-Guided Bronchoscopy: Feasibility and Detection Depth Studies Using Ex Vivo Lung Tissues and SERS Nanoparticle Tags

**Zongyu Wu [1], Ziwen Wang [2], Haoqiang Xie [1], Yiming Wang [1], Haoqi He [1], Shuming Nie [2], Jian Ye [1] and Li Lin [1,***

[1] School of Biomedical Engineering, Shanghai Jiao Tong University, Shanghai 200030, China; 2020wzy@sjtu.edu.cn (Z.W.); xiehaoqiang@sjtu.edu.cn (H.X.); wym-flpb@sjtu.edu.cn (Y.W.); hehaoqi@sjtu.edu.cn (H.H.); yejian78@sjtu.edu.cn (J.Y.)
[2] Department of Bioengineering, University of Illinois at Urbana-Champaign, Urbana, IL 61801, USA; ziwenw3@illinois.edu (Z.W.); nies@illinois.edu (S.N.)
\* Correspondence: linli92@sjtu.edu.cn

**Abstract:** Image-guided and robotic bronchoscopy is currently under intense research and development for a broad range of clinical applications, especially for minimally invasive biopsy and surgery of peripheral pulmonary nodules or lesions that are frequently discovered by CT or MRI scans. Optical imaging and spectroscopic modalities at the near-infrared (NIR) window hold great promise for bronchoscopic navigation and guidance because of their high detection sensitivity and molecular/cellular specificity. However, light scattering and background interference are two major factors limiting the depth of tissue penetration of photons, and diseased lesions such as small tumors buried under the tissue surface often cannot be detected. Here we report the use of a miniaturized Raman device that is inserted into one of the bronchoscope channels for sensitive detection of "phantom" tumors using fresh pig lung tissues and surface-enhanced Raman scattering (SERS) nanoparticle tags. The ex vivo results demonstrate not only the feasibility of using Raman spectroscopy for endoscopic guidance, but also show that ultrabright SERS nanoparticles allow detection through a bronchial wall of 0.85 mm in thickness and a 5 mm-thick layer of lung tissue (approaching the fourth-generation airway). This work highlights the prospects and potential of Raman-guided bronchoscopy for minimally invasive imaging and detection of lung lesions.

**Keywords:** bronchoscopy; Raman spectroscopy; endoscopic Raman probe; fluorescence; detection depth

## 1. Introduction

Lung cancer has long been the leading cause of cancer-related mortality. Although open surgery is still the mainstream of lung cancer resection, bronchoscopy intervention has become a preferable clinical procedure to sample the pulmonary lesions and perform minimally invasive surgeries, with lower complication rates and reduced airway bleeding [1]. So far, the bronchoscopy navigation has reached up to ninth-generation bronchi with the newly emerging robotic platform, enhancing the ability to mark and diagnose small lesions at high generations [2–4]. To better locate the peripheral pulmonary lesions, advanced bronchoscopic guidance techniques have been developed, including endobronchial ultrasonography, electromagnetic (EM) navigation and computed tomography (CT) image-based virtual bronchoscopy navigation [5–7]. Optical imaging and spectroscopic modalities, as widely used in intraoperative guidance, hold great promise for bronchoscopic navigation because of their high detection sensitivity and molecular/cellular specificity [8]. The introduction of image-guided bronchoscopy is anticipated to aid with the precise sensing of lesion tumors.

As one of the emerging optical detection modalities, surface-enhanced Raman spectroscopy (SERS) has attracted abundant attention in the fields of biology and medicine. SERS stands out for its ultrahigh sensitivity, favorable photostability and narrow linewidth, which reduces the interference from tissue autofluorescence background and benefits multiplexing detection ability [9,10]. With a variety of SERS nanoparticle (NP) tags being developed as the optical labels for imaging and biosensing [11–14], SERS-guided bronchoscopy is therefore highly anticipated.

However, light scattering and background interference are two major factors limiting the depth of tissue penetration of photons, and diseased lesions such as small tumors buried under the tissue surface often cannot be detected [15]. Conventional ultrasound and CT imaging have detection depths of 15–20 cm or more in patients, allowing three-dimensional lung scanning [16–18]. In comparison, optical detection remains relatively shallow, with the mean free path of a photon in the order of 100 μm in bio-tissues [19]. During bronchoscopy, the Raman endoscopic probe is placed inside the airway, and the lung lesions are mainly distributed outside in the tissue interstitium, either adjacent or non-adjacent to the airway wall. Therefore, questions remain as whether the optical signal can penetrate the bronchi wall and lung tissues to detect lesions during non-invasive SERS-guided bronchoscopy (Figure 1). Ex vivo studies allow researchers to flexibly measure intensities in an experimentally controlled manner, providing a general answer to photon propagation in tissues before translating into a complex physiological environment. Therefore, ex vivo investigations of SERS detection depths are critical to promote the clinical translation of Raman modalities in the diagnosis of lung cancers. Nevertheless, to the best of our knowledge, the feasibility of using SERS for endoscopic guidance has not been proven so far, and the quantitative assessments of SERS detection depths in lung tissues are still lacking.

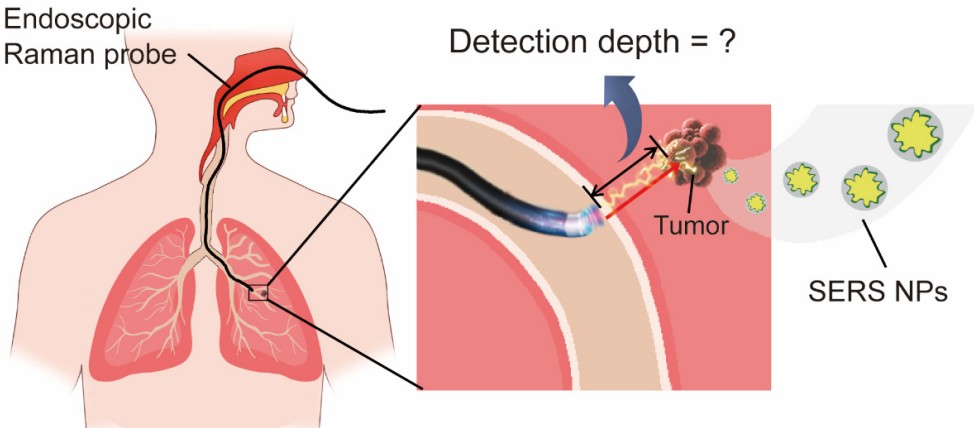

**Figure 1.** Schematic diagram of Raman-guided bronchoscopy in detecting lung tumors with the use of SERS NPs.

In this work, we have demonstrated the feasibility of SERS-guided bronchoscopy and quantitatively assessed its detection depth in ex vivo lung tissues. This was fulfilled with the use of a miniaturized Raman endoscopic probe that could be inserted into one of the bronchoscope channels for sensitive detection of "phantom" tumors embedded in pig lung tissues. We synthesized gold–silver (Au-Ag) bimetallic SERS NPs and prepared agarose gels tagged with SERS NPs (termed as SERS gels) as the phantom lesions. The ex vivo results demonstrate not only the feasibility of using Raman spectroscopy for endoscopic guidance, but also show that ultrabright SERS NPs allow detection through a bronchial wall of 0.85 mm in thickness and a 5 mm-thick layer of lung tissue (approaching the fourth-generation airway). Our assessments were also applicable to fluorescence spectroscopic imaging. The fluorescence detection depth of phantoms labeled by an FDA-approved fluorophore (indocyanine green, ICG) was measured on a commercial image-guided surgery system. Its valid detection depth is 4 mm in lung tissues without the bronchial wall. This work provides a fundamental understanding of the detection

depth of SERS in tissues, showing the possibility of SERS and fluorescence modalities being integrated for a better diagnosis of pulmonary lesions. Furthermore, it highlights the potential of the SERS-guided bronchoscopic platform in intraoperative diagnostics.

## 2. Methods and Materials

### 2.1. Materials

Chloroauric chloride ($HAuCl_4 \cdot 4H_2O$), ethanol ($\geq$99.7%), and N,N-dimethylformamide (DMF, $\geq$99%) were received from Sinopharm Chemical Reagent Co. Ltd. (Shanghai, China). Cetyltrimethylammonium chloride (CTAC, 99%) and sodium borohydride ($NaBH_4$, 98%) were purchased from J&K Chemical Ltd. (Shanghai, China). Silver nitrate ($AgNO_3$, 99.8%) and ascorbic acid (>99.0%) were obtained from Aladdin (China). 4-Nitrobenzenethiol (4-NBT) and IR-780 iodide (IR-780, 98%) were acquired from Sigma-Aldrich (Shanghai, China). Nanopure water (18.2 M$\Omega$) was used for all experiments. Bovine serum albumin (BSA), dimethyl sulfoxide (DMSO), and ascorbic acid were purchased from Sigma-Aldrich (St. Louis, MO, USA). Phosphate-buffered saline (PBS) was purchased from Corning (Corning, NY, USA). Indocyanine green (ICG) was purchased from TCI America (Portland, OR, USA). Agarose was purchased from Fisher Scientific (Hanover Park, IL, USA). All materials were used as received without any further purification.

### 2.2. Preparation of SERS NPs and SERS Gels

SERS NPs were fabricated in two steps according to our reported protocols [20,21]. As shown in Figure 2, first, the petal-like Au NPs (40~50 nm) were synthesized by slightly modifying the previously published protocols [22,23]. Then, 6 mL of Au NPs (0.5 nM) were used as cores and mixed with 1 mL of IR-780 Raman reporter molecules (0.32 mM, dissolved in DMF). After 2 h, the mixture was centrifuged to remove excess IR-780 molecules. Amounts of 1.8 mL of $AgNO_3$ (14.6 mM) and 4.5 mL of ascorbic acid (100 mM) were added to the above molecule-modified Au NPs and the mixture was incubated for 2 h at 70 °C to form the Au/Ag bimetallic SERS NPs. After washing and being redispersed in the aqueous solution, the final concentration of SERS NPs was set as 0.25 nM. The morphologies of NPs were confirmed using TEM images, collected from a JEM-2100F transmission electron microscope (JEOL, Tokyo, Japan) operated at 200 kV. The UV–Vis extinction spectra of NPs were measured from a UV1900 UV–Vis spectrophotometer (Aucybest, Shanghai, China).

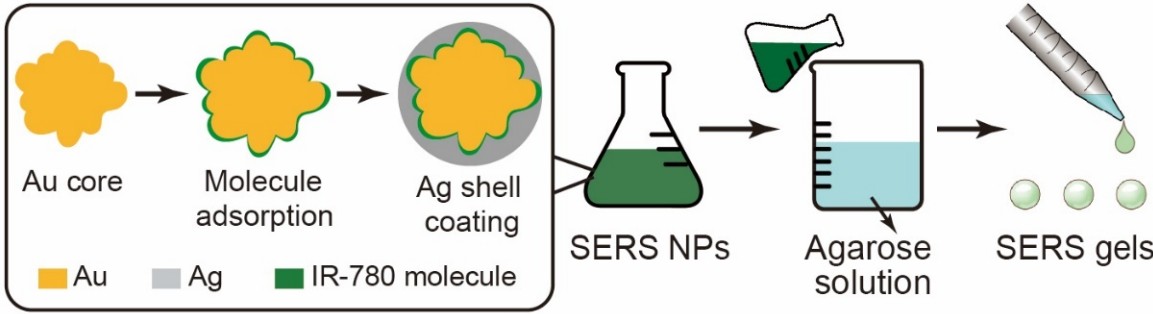

**Figure 2.** Preparation scheme of SERS NPs and SERS gels.

With the obtained SERS NPs, we continued to prepare the SERS gels (Figure 2). Briefly, 1 wt% agarose was mixed with water and heated in a microwave oven until all of the agarose was fully dissolved. Then the agarose solution and the SERS NPs were mixed in the ratio of 9:1 or 99:1, to obtain NP concentrations of 25 or 2.5 pM. Under stirring, the SERS NP colloids were well-dispersed in heated agarose solution. The mixture was kept in a boiling-water bath to avoid solidification. Then, we utilized a pipette to extract a drop from the mixture, which was dropped onto a glass substrate to solidify and form a spherical bead, i.e., the SERS gel.

### 2.3. SERS Measurements on Ex Vivo Tissues and Spectral Analysis

The SERS measurements were carried out using a miniatured Raman fiber optic probe and SERS gels as the lesion phantom (as shown in Figure 3). The endoscopic Raman probe (Emvision, Loxahatchee, FL, USA) is 2.1 mm in diameter and 3 m in length, coupled with a 785 nm laser and a high-resolution spectrometer (Andor, London, UK), as reported in our previous study [24]. The laser spot is ~300 μm in diameter. Figure S1a displays the Raman spectrum of the background (i.e., measured without any samples). There are three relatively broad peaks at 486, 617 and 819 cm$^{-1}$, which can be attributed to optical fiber materials. These are the Raman signals generated in the silica fiber optic cables during the delivery of light. We call it the background of the device.

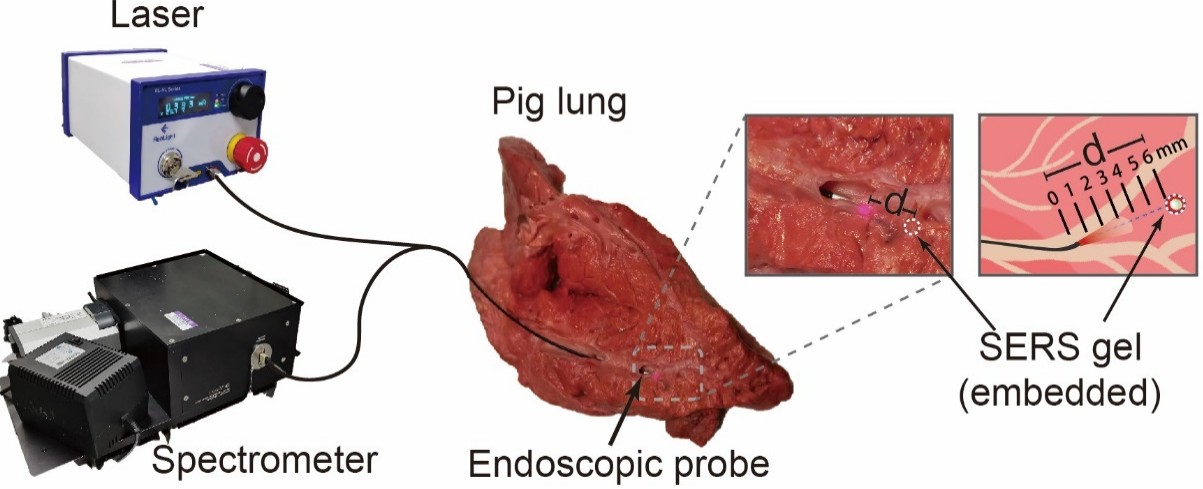

**Figure 3.** Raman assessments on ex vivo lung tissues using the fiber optic probe.

The ex vivo assessments of SERS detection depths were carried out using a miniatured Raman fiber optic probe and SERS gels as the lesion phantom. A fresh pig lung was purchased from a local grocery (Figure S2). We used a knife to cut the lung in half and make a transverse section where the bronchus cross section could be seen. During the Raman endoscopy mimetic experiments, the fiber optic probe was placed in the bronchus airway and the probe contacted the bronchi wall directly; the other end of the probe was connected to the laser and Raman spectrometer (Figure 3). We selected sites with relatively homogeneous lung tissue to conduct the assessments. The bronchus wall thickness for each site was measured using a vernier caliper. The SERS gel was embedded in the lung tissues at a distance (*d* = 1–6 mm) from the bronchi wall. A vernier caliper was used to measure the distances to mark 6 distance points. The Raman measurements were carried out with the gel at different distances. To make sure the change of tissue structure where the gel was inserted would not bias the results, we started from a high distance, i.e., following the order of distances from 6 mm to 1 mm. After each measurement, we took the SERS gel out and carefully sealed the tissue to ensure no air or fluid gaps. A Raman test was also conducted to ensure there was no residual SERS gel before placing a new one at the next distance point. To sum up, we selected 6 different locations in the lung to conduct the assessments; for each location, 6 distances (1–6 mm) were investigated and for each distance the Raman spectra were collected 3 times. The laser power (20 mW) and integration time (5 s) remained the same during the measurements.

The baselines of raw spectra were corrected using a wavelet transform algorithm to remove the tissue autofluorescence background [25]. Raman signals were then evaluated based on the spectral signal-to-noise ratio (SNR), measured as the ratio of the peak height over the standard deviation of the noise floor. For an effective spectrum, an SNR of 3 (3σ)

is minimally acceptable with a confidence level at over 99% for the peak not being random noise [26]. This can be mathematically described by the equation:

$$SNR = \frac{I_{signal}}{\sigma_{noise}} \geq 3 \qquad (1)$$

We chose one of the strongest Raman modes (peak at 1206 cm$^{-1}$) of IR-780 reporter molecules to calculate $I_{signal}$, since this peak was not overlapped with the device or tissue background (Figure S1b,c). The peak height was calculated by averaging the data in the range of 1203–1209 cm$^{-1}$. As for the calculation of $\sigma_{noise}$, we chose a region of 1950–2050 cm$^{-1}$, because it was in the biological Raman-silent spectral window (1800–2700 cm$^{-1}$) [27], and there were no Raman peaks from lung tissues, the device, or IR-780 molecules (Figure S1).

*2.4. Fluorescence Detection Depth Measurement and Data Analysis*

For the assessments of fluorescence detection depth, agarose gels tagged by FDA-approved ICG fluorophore were used as the phantom lesions (named as ICG gels). We prepared a series of ICG gels with different fluorophore concentrations. Briefly, ICG was first dissolved in DMSO to obtain 1 mM stock solution. It was then diluted in PBS 1X with 0.1 mM BSA to yield a series of ICG solutions. Again, 1 wt% agarose was fully dissolved in water under heating. The agarose solution and ICG solutions were mixed in a 9:1 ratio and cured in a standard 12-well culture plate at room temperature. This finally produced gels with ICG concentrations of 10 nM, 100 nM, 1 μM and 10 μM. We confirmed that the fluorescence intensity of ICG exhibited almost no change during the gel preparation process (Figure S3). An image-guided surgery system (REAL-IGS, LIFENERGY Nanjing Nuoyuan Medical Devices, Nanjing, China) was utilized to capture the fluorescent images. The imaging mode was equipped with a 785 nm planar laser, which illuminated an area of 10 cm × 10 cm with a working distance of 50 cm from the laser probe to the samples. To avoid potential photobleaching, a relatively low power density (4.8 mW/cm$^2$) was adopted, with the laser output power at 480 mW. The acquisition time (492 μs) and CCD gain (592) were selected, which were the maximum values of the instrument, in order to explore the limit of fluorescence detection depth.

The fresh lung tissues were frozen before use. A meat slicer was applied to cut the frozen lung tissues into thin layers with a thickness of 1.5 to 5 mm. A vernier caliper was used to measure the thickness of each slice. During the measurements, the lung tissues (without bronchi walls) were put onto and fully covered the ICG gel. The fluorescence image was captured from the top. The intensity of each image was measured from the selected region of interest (ROI), which was slightly smaller than the gel area in the image to eliminate the glowing edges. The average intensity in each image was calculated and plotted to give quantitative information about the effects of tissue thickness and ICG concentration on fluorescence. Each image was processed in MATLAB with pseudocolor. The SNR of the fluorescence image was defined as the difference between the average fluorescence intensities of the sample ($I_{signal}$) and the intensity standard deviation ($\sigma_{noise}$) of the background. The pure agarose gel without ICG was adopted as the blank sample, whose image intensity deviation was applied as the background noise ($\sigma_{noise}$).

**3. Results**

To begin with, we have carefully synthesized SERS NPs and SERS gels. SERS NPs were fabricated in two steps: firstly, the Au NPs with a flower-like rough surface were synthesized; then the NPs were coated with a layer of Raman reporter molecules (IR-780) and the Ag layer, forming the bimetallic NPs with a smooth surface (Figure 2). IR-780 is a commercial fluorophore dye with an absorption peak close to 785 nm (Figure S4), serving as a resonant Raman reporter molecule. The transmission electron microscopy (TEM) images in Figure 4a confirm the rough surface and smooth structure before and after Ag coating, respectively. The final product is 70–80 nm in diameter. In Figure 4b, the

UV–Vis spectrum of Au cores shows a resonant peak at around 608 nm, which is a typical peak of Au colloids with a rough surface. For the UV–Vis spectrum of Ag-coated NPs, the peak at 450 nm corresponds to the resonance position of Ag nanomaterials, confirming the successful coating of the Ag shell. With these obtained SERS NPs, we continued to prepare the SERS gels (Figure 4c). The diameter of the gel beads was ~400 μm, measured by a vernier caliper. SERS gels were prepared with NP concentrations of 25 or 2.5 pM. In this way, we could attain the Raman-active material in a solid state while maintaining the uniform dispersion of NPs.

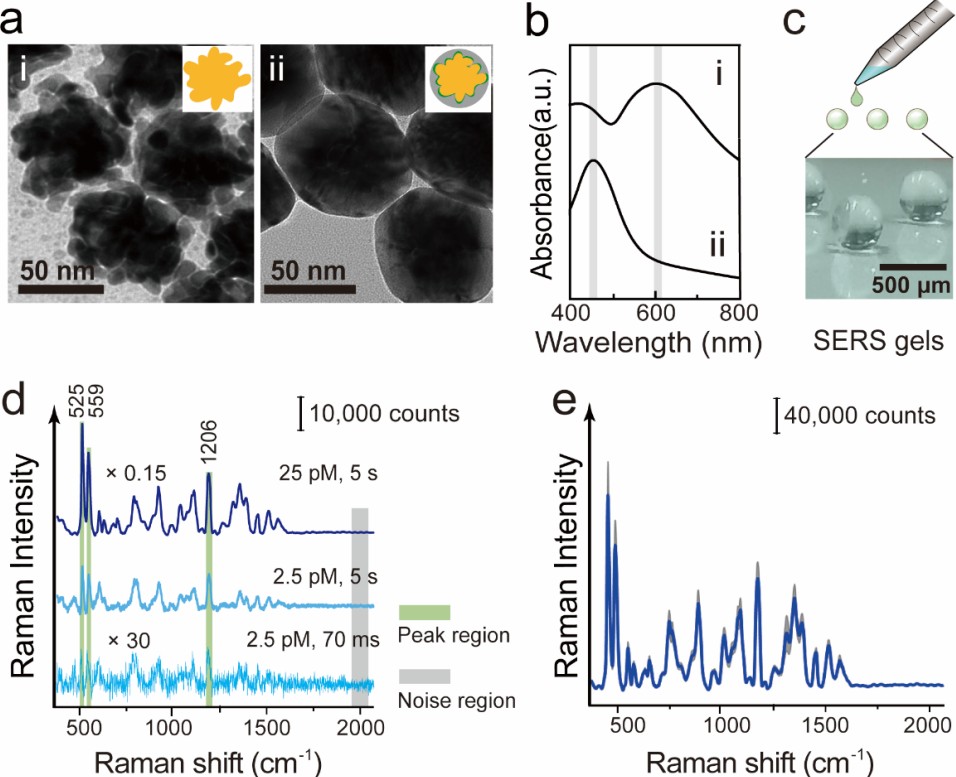

**Figure 4.** The procedures for SERS gel preparation and Raman measurements. (**a**) TEM images and (**b**) extinction spectra for (**i**) petal-like Au cores and (**ii**) Ag-shell-coated SERS NPs. (**c**) The preparation of SERS gels. (**d**) Raman spectra of SERS gels containing 25 or 2.5 pM NPs. Raman peaks of the gels (525, 559, 1206 cm$^{-1}$) are marked in green, and the region for noise deviation calculation (1950–2050 cm$^{-1}$) is marked in grey. The excitation power was 20 mW. (**e**) Measurements on three SERS gels containing 25 pM NPs. The measurements were repeated 30 times. The data were plotted using averages and the shadows for standard deviations.

We then measured the Raman response of the SERS gel to determine its characteristic peaks and LoD. The fiber optic Raman probe was utilized to excite and detect signals under 785 nm excitation. As shown in Figure 4d (top), prominent Raman signals are obtained for SERS gels containing 25 pM NPs, with 20 mW laser power and 5 s acquisition time. The gel exhibits three strong bands at 525, 559 and 1206 cm$^{-1}$, corresponding to phenyl out-of-plane substituent-sensitive vibration, phenyl in-plane deformation, and C-H in-plane deformation of IR-780 molecules [28]. We define the spectral SNR as the ratio of signal intensity ($I_{signal}$) at the Raman peak of 1206 cm$^{-1}$ to the standard deviations ($\sigma_{noise}$) of the noise floor. The SNR of 25 pM SERS gel (Figure 4d, top) is calculated to be 297. For our NPs, IR-780 Raman reporter molecules are hidden inside the Ag shell; this unique structure protects the molecules from desorption or interference with aggregates, rendering excellent signal stability and uniformity [21]. To confirm this, we repeatedly measured the Raman spectra from three SERS gels. The gels were placed onto a quartz substrate during the measurements; we pointed the Raman probe directly towards a bead and collected the

signals. As shown in Figure 4e, the spectra show very good uniformity and batch-to-batch reproducibility, with the SNR standard deviation being only ~4.8%.

We continued to measure SERS gels of 2.5 pM NPs using an integration time of 5 s; its spectral SNR is 48 (Figure 4d, middle). If further decreasing the acquisition time until it reaches the lower limit of our Raman spectrometer, i.e., 70 ms in this case, we can still observe an effective spectrum with the SNR of more than 3 (Figure 4d, bottom). This result indicates the LOD of SERS gels is down to 2.5 pM in concentration and 70 ms in acquisition time. The ultrahigh brightness and low LOD of SERS gels open the possibility of reducing the dosage or integration time when using SERS NPs as imaging tracers in real clinical situations. The gel can serve as a phantom lesion tagged by NPs for our quantitative assessments on SERS detection depth in ex vivo lung tissues.

Before the assessments, we also evaluated whether the NP dose of the SERS gel was within a reasonable range. When injected intravenously, NPs are delivered into solid tumors through a process of circulation, accumulation, penetration and internalization. The NP delivery efficiency was reported to be at a median of 0.7% of the injected dose [29]. To mimic tumor lesions, the key point is that the NP quantity within the "phantom" should not be much higher than that of a real lesion tagged by NPs. This could be evaluated either in concentration or in amount. Our SERS gels contain NPs of 2.5–25 pM, which corresponds to $1.5–15 \times 10^9$/mL; as the gel is 400 μm in diameter, the NP amounts are $0.5 \times 10^5–10^6$ per gel. We compare this value to an initial study of a clinical trial in humans by N. Halas et al. [30]. They found that the mean concentration of Au NPs in sampled core tumors was in the range of $0.078–2.26 \times 10^9$/mL; our NP concentrations are close to this range. Also, we refer to a recent report by Chan et al., who demonstrated that NP tumor delivery was dependent on the injected amount [29]. They used an injection dose of $3 \times 10^{10}–5.5 \times 10^{14}$ NPs, with the delivery rate ranging from 0.03–12%; this would translate into around $9 \times 10^6–6.6 \times 10^{13}$ NPs available at the tumor. The NP amounts in our SERS gels are slightly below their results of a low-dose injection. Therefore, we assume that our gels, regarding either the concentration or the number of NPs, are within the safe and appropriate range to mimic the NP-tagged lesions. Also, the SERS gels are quite small, on a micro-scale. This allows them to well mimic microtumors.

We set out to assess the SERS detection depths as the gels were embedded at different distances from the airway. The SERS gels with 25 pM NPs were first utilized as the phantom lesions. We selected three sites at around second–fourth-generation bronchi for evaluations (the scheme of generations can be found in Figure S5), since these generations can be easily reached in current navigation bronchoscopy [4]. Figure 5a displays two representative sites (site **i** and **ii**). To excite and collect signals from the SERS gel, the endoscopic Raman probe directly touched the internal bronchi wall and pointed in the direction of the SERS gel. After each measurement, the SERS gel was taken out and a new one was placed into the decreasing distance point. The collected Raman spectra for site **i** and **ii** at different distances (1–6 mm) are presented for a direct comparison (Figure 5b). Overall, the Raman intensity of SERS gels at 525, 559 and 1206 cm$^{-1}$ decreases as the distance increases. Another characteristic peak at 1450 cm$^{-1}$ is attributed to lung tissues, corresponding to the CH$_2$ deformation peak [31]. When the distance increases to 6 mm, the collected Raman signal appears the same as that of pure lung tissues (Figure S1d), indicating the SERS gel is not detectable at this distance. The Raman modes of SERS gels at 525 and 559 cm$^{-1}$ are slightly overlapped by the background. Therefore, we choose the peak height of 1206 cm$^{-1}$ ($I_{signal}$) to calculate SNR. SERS detection depths are quantitatively defined as the maximal distance at which an effective signal of the SERS gel could be recognized with the SNR of ≥3. For site **i**, where the bronchi wall is of 0.85 nm in thickness, the detection depth is the highest (5 mm); for site **ii** with its bronchi wall of 1.06 mm, the maximal distance is 4 mm. If moving the test site to the second-generation bronchus, the airway wall increases to 1.29 mm, and the detection depth decreases to 3 mm (site **iii**, see Table 1). This result is reasonable considering that the bronchi wall is a scattering medium of light. Our measured

detection depths are affected by many factors including measurement conditions, lung tissue attenuation, the brightness of SERS gels and so on.

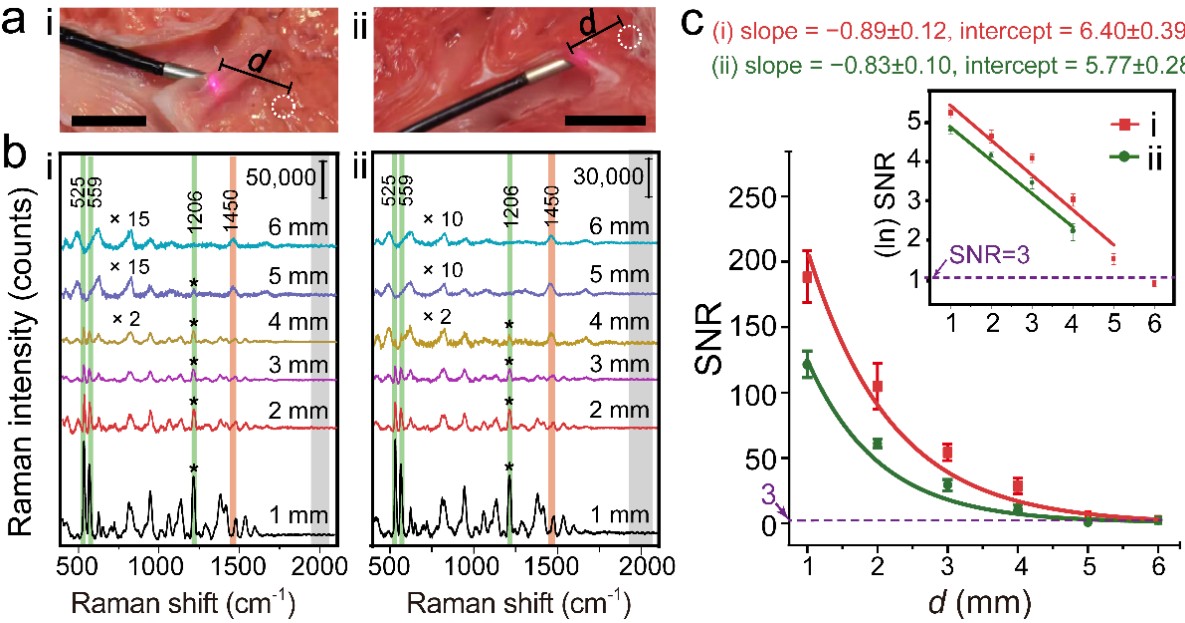

**Figure 5.** Quantitative assessments of Raman spectroscopic detection depth in lung tissues. (**a**) Photographs of measurements at (**i**) site **i**, and (**ii**) site **ii**. The white dashed circle indicates the location of a SERS gel at a distance (**d**) to the bronchi wall. (**b**) The Raman spectra and (**c**) SNR profile of SERS gels embedded in lung tissues at distances (**d**) from 1 to 6 mm. In (**b**), Raman peaks of SERS gels (525, 559 and 1206 cm$^{-1}$) are marked in green, peak of lung tissues (1450 cm$^{-1}$) is marked in orange, and the region (1950–2050 cm$^{-1}$) for noise calculation is marked in grey. The measurements were performed under 785 nm excitation (20 mW, 5 s integration time). The inset in (**c**) is plotted in an ln Y-axis, and parameters of linear fitting models are displayed, with R-squared values of 0.93 and 0.96 for the red and green line, respectively.

**Table 1.** The thickness of bronchi wall, SERS NPs concentrations, the corresponding detection depths as well as SNR of peak 1206 cm$^{-1}$ at each site.

| Site | Thickness of Bronchi Wall | Concentration of SERS NPs | Detection Depth in Lung Tissues | SNR at the Detection Depth |
|------|---------------------------|---------------------------|----------------------------------|----------------------------|
| i    | 0.85 mm                   | 25 pM                     | 5 mm                             | 4.6 ± 0.40                 |
| ii   | 1.06 mm                   | 25 pM                     | 4 mm                             | 9.5 ± 2.10                 |
| iii  | 1.29 mm                   | 25 pM                     | 3 mm                             | 28.4 ± 3.11                |
| iv   | 0.76 mm                   | 2.5 pM                    | 2 mm                             | 11.0 ± 2.32                |
| v    | 0.80 mm                   | 2.5 pM                    | 2 mm                             | 4.8 ± 0.22                 |
| vi   | 1.22 mm                   | 2.5 pM                    | <1 mm                            | /                          |

To provide a general determination of detection depths or the system's ultimate sensitivity, we investigated the properties of lung tissues and plotted the relationship between SNR and the distance (tissue thickness) in Figure 5c. The decrease in spectral SNR follows an exponential decay regarding tissue thickness, according with the existing studies on light–tissue interactions [32]. The relationship can be described as:

$$SNR_d = SNR_0 e^{-\sigma d} \tag{2}$$

where *d* is the distance from the SERS gels to the bronchi wall. $SNR_d$ represents the spectral SNR measured at *d*. $SNR_0$ is the original SNR at *d* = 0; this value is affected by SERS gel brightness, bronchi wall thickness, integration time and laser power. $\sigma$ describes the

attenuation properties of lung tissues and is the only parameter to determine the decay rate of SNR. Equation (2) can also be described as:

$$\ln(SNR_d) = -\sigma d + \ln(SNR_0) \tag{3}$$

Based on Equation (3), the data were further plotted using an ln Y-axis (see the inset of Figure 5c). The points with an SNR of over 3 are clearly displayed and can be fitted using linear models. The fitting was performed on a limited number of points because we were only able to get a step of 1 mm in tissue thickness, restricted by the current experimental technique. Still, we could have some interesting findings based on the present data. A smaller intercept is observed for site **ii** (5.77 ± 0.28) compared with that of site **i** (6.40 ± 0.39); this is reasonable since a thicker bronchi wall at site **ii** results in a lower $SNR_0$. Meanwhile, the slope $(-\sigma)$ of the line is $-0.89 \pm 0.12$ for site **i**; this corresponds to the value of $e^{-\sigma}$, ranging from 0.37 to 0.46. The line slope for site **ii** is $-0.83 \pm 0.10$, corresponding to an $e^{-\sigma}$ of 0.39–0.49. Based on this, we could make a rough estimation that the SNR decreases in the range of 37–49% $(e^{-\sigma})$ as the distance increases by 1 mm. These are the intrinsic properties of lung tissues and would be generally applicable for other SERS NPs with different concentrations or other systems. In other words, with a given $SNR_0$, the slope value would allow the prediction of the detection depths. For example, we have made an estimation on SERS gels with a lower concentration (2.5 pM) of NPs. As shown in Figure 4d, the SNR of 2.5 pM SERS gel is 0.162 times that of the 25 pM SERS gel. Assuming the same thickness of bronchi wall and the same measurement conditions, the SNR at the distance of 0 for the 2.5 pM gel is

$$SNR_0' = 0.162 \times SNR_0 \tag{4}$$

Since their detection depth threshold is the same (SNR = 3), then we have:

$$SNR_0 e^{-\sigma d_1} = SNR_0' \, e^{-\sigma d_2} \tag{5}$$

where $d_1$ and $d_2$ is the detection depth for 25 pM and 2.5 pM SERS gels, respectively. Combining (4) and (5), we get

$$d_1 - d_2 = \frac{\ln(0.162)}{-\sigma} \tag{6}$$

It can be calculated that the detection depth difference between 25 pM and 2.5 pM SERS gel is roughly in the range of 2–3 mm. To confirm this estimation, we have experimentally tested the 2.5 pM SERS gels on three sites (sites **iv**–**vi**, see Table 1). Site **iv** and **v** are near the fourth-generation bronchi with a bronchi wall of 0.76–0.80 mm; their detection depths are both 2 mm. When moving to the second-generation airway with a 1.22 mm-thick bronchi wall, the detection depth decreases to less than 1 mm. If we compare site **i** and **v**, which have similar thicknesses of the bronchi walls, their detection depth difference is ~3 mm; similarly, if comparing site **iii** and **vi**, the depth difference is 2–3 mm. These experimental results match well with our estimation. To sum up, we achieved detection depths of up to 5 mm on SERS gels of 25 pM NPs around the fourth-generation airways. Also, we provided general information on lung tissue properties, making it possible to estimate detection depths with a known $SNR_0$. The results reveal the possibility of using SERS-based bronchoscopy for the sensing of small pulmonary lesions.

Our assessments of detection depth in ex vivo tissues are also feasible for fluorescence. Since fluorescence has been widely used to offer real-time visualizations of a large area during open surgeries on lung cancers (Figure 6a), the ex vivo studies of its detection depth offer a significant reference for clinical practices. ICG gels were prepared as phantom lesions. To mimic the fluorescence scanning of lung surgeries, during which the lung is cut open and the laser can penetrate lung tissues directly to reach the lesion, we covered the ICG gel with lung tissues (1.5–5 mm in thickness) without the bronchi wall. An image-guided surgery system was placed on top to capture the fluorescence image of tissue-covered

gels (Figure 6b). ICG gels were prepared with the concentration to be 10 nM, 100 nM, 1 μM or 10 μM (Figure 6c). When covered by lung tissues, the overall trend of intensity decay over increasing tissue thickness is revealed (Figure 6d). This relationship can also be well-described in exponential decay. The fluorescence detection depth is defined as the largest thickness of lung tissues with which an image with an SNR over 3 could be captured from the tissue-covered ICG gel. All the detection depths are recorded in Table S1. The highest detection depth is 4 mm for a 1 μM ICG gel, and it decreases to 3 mm for a 10 μM gel. The failure to show improvements in detection depth for 10 μM gel could be attributed to the concentration quenching and higher photobleaching rate. As for the 100 nM gel, a fluorescence image can be captured with 2 mm lung tissues. The 10 nM gels are not able to be detected with the lung tissues on top.

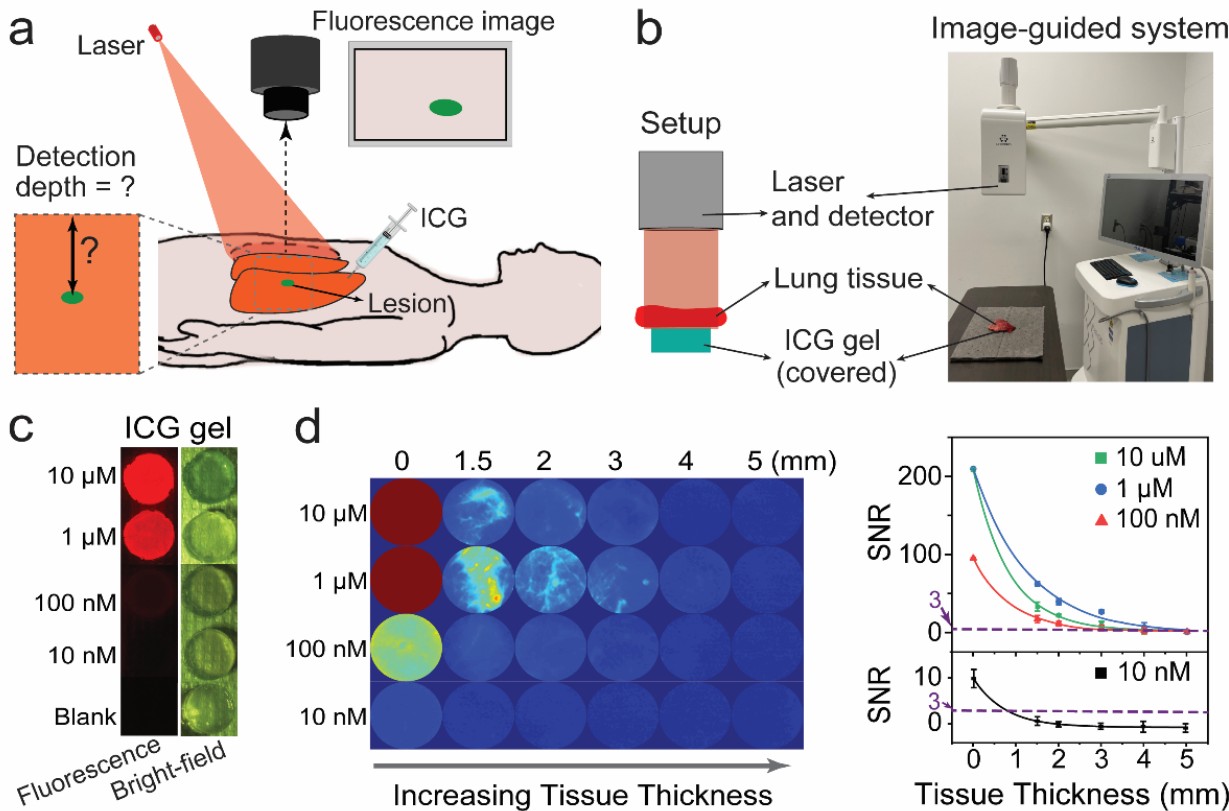

**Figure 6.** Measurements of the detection depth of fluorescence imaging and spectroscopy in lung tissues. (**a**) Schematic of NIR fluorescence imaging on lung cancer patients. (**b**) The experimental setup. (**c**) Recorded bright and fluorescent images of ICG gels (10 μM, 1 μM, 100 nM, 10 nM). (**d**) Fluorescent images and corresponding SBR of ICG gels covered by lung tissues of different thicknesses (0–5 mm).

## 4. Discussion

We demonstrate on "phantom" tumors that SERS-guided bronchoscopy achieves detection through up to 5 mm lung tissues and 0.85 mm bronchi walls. This benefits from the ultrahigh sensitivity of our SERS NPs. Our investigations into the optical properties of lung tissues further reveal the possibility to estimate detection depths for different NPs with different concentrations. Although more studies in complex clinical environment are needed, the results of the present experiments set the stage for a clinical study of SERS-guided bronchoscopy in intraoperative diagnosis and imaging.

Together with fluorescence imaging, the incorporated techniques allow not only the detection of ultra-small microsurgical sites on patients, but also a straightforward wide-field visualization of tumor regions, to achieve a better diagnosis of pulmonary micro-tumors. The assessment of fluorescence detection depth is meaningful because fluorescence is the

modality that SERS could be integrated with. While we should note that our assessments on SERS and NIR fluorescence depth sensitivity are not a direct comparison because the instruments and data acquisition conditions are very different. In fact, the measured depth sensitivity via ICG is valid only for the utilized measurement device in this work. In fact, if a more sensitive imaging system or fiber probe was utilized, the detection depth would likely be higher. For example, ICG was detected to 12 mm in a deflated lung (equivalent to ~24 mm in an inflated lung) using a "bead insertion" phantom, similar to the one here [33].

Whether a detection depth of 5 mm in lung tissues is enough for routine clinical practice remains controversial, considering the complex bio-tissue environments. Yom et al. have reported a quantitative way to describe the lung lesion proximity to the bronchial tree [34]. In routine clinical practices, the tumor location is defined as central if any portion of the tumor margin is at the boundary of the proximal bronchial tree zone. For these central tumors, the minimum distance between the airway and tumor could range from <1 mm up to a few centimeters [34]. However, the newly emerging robotic platform has shown the ability to detect small lesions with farther access to the periphery of the lung, such as the ninth airway generation [35]. As the airway generation increases, the bronchi diameter decreases, as well as the distances between two airway branches (Figure S5). The bronchial walls get thinner too, with a thickness of ~20% of the bronchial luminal diameter [36]. On one hand, this benefits the lesion detection of SERS-guided bronchoscopy, since a lower detection depth is adequate in such a case; on the other hand, this brings new challenges as there is a demand for the development of thinner fiber optic probes to approach higher airway generations.

A future study would be in demand to determine the optimal combination of integration time, laser density and injection dose, to lead to the best sensitivity for the specific detection. First, the extension in integration time improves the detection performance by collecting more Raman photons. SERS NPs generally exhibit better photostability than organic fluorophores [37], allowing us to use a longer integration time. However, the integration time should not be too long to make the clinical procedure inconvenient. It is therefore not practical to increase SNR by extending the integration time. Second, a wide range of applied laser powers, from 10 to 150 mW, has been reported in the current literature for Raman optic-fiber-probe-based in vivo detections [38–41]. Our laser power is reasonably within this range. To further lower the laser power density in practical applications, we believe that a collimated illumination laser, instead of a common Gaussian beam, could be used [40]. This could help to reliably keep the power density below maximum permissible exposure (MPE) limits (as defined by ANSI-2000). In addition, the second near-infrared (NIR-II) window excitation exhibit the merits of higher MPE compared with that of the NIR-I window, due to the lower tissue absorption and lower energy of photons at longer wavelengths [42]. Therefore, SERS NPs at the NIR-II window could be designed to better satisfy the requirement of MPE in the future. Last, it is possible for us to further improve the detection depth by increasing the NPs accumulated at the tumor. As discussed before, the NP amounts of our SERS gel are still below that of a real case. For practical usage, we can boost NP tumor delivery by increasing the injection dose within a suitable range, to enhance the Raman intensity of NP-tagged tumors. Also, we would like to note that, although Ag is not usually regarded as a biocompatible material, our NPs can be encapsulated by biocompatible polymer or silica layers to improve its safety for biomedical applications [20].

## 5. Conclusions

We investigated the potential and feasibility of using Raman spectroscopy for endoscopic guidance on phantom tumors. We first fabricated the ultrabright SERS NPs and made them into SERS gels that exhibited a detection limit of down to 2.5 pM with 70 ms acquisition time, presenting the promise of low dosage and short acquisition time in future applications. The SERS gels served as phantom lesions and were embedded in ex vivo lung tissues. Our assessment results demonstrated successful detection through a bronchial wall

of 0.85 mm in thickness and a 5 mm-thick layer of lung tissue (approaching the fourth-generation airway). Furthermore, the analysis of attenuation properties in lung tissues revealed the possibility to estimate the detection depths of other SERS NPs with different concentrations. We have also assessed the fluorescence detection depths using a commercial image-guided surgery system; it was shown to detect phantom lesions (labeled with ICG) through 4 mm lung tissues without the bronchial wall, with the ICG concentration of 1 μM. The endoscopic Raman probe can flexibly fit into the ultra-small surgical sites in patients and fluorescence imaging offers a straightforward visualization of tumor regions. The combination of both modalities can effectively adapt to the complex clinical environment. This work sets the stage for an in vivo clinical study and highlights the prospects of a SERS-guided bronchoscopic platform in intraoperative diagnostics.

**Supplementary Materials:** The following supporting information can be downloaded at: https://www.mdpi.com/article/10.3390/photonics9060429/s1, Figure S1: The Raman spectra of (a) device background (collected without any samples), (b) lung tissues, (c) a SERS gel containing IR-780 as the Raman reporter, and (d) the SERS gel embedded at site i of the lung tissue, with tissue thickness of 6 mm. Figure S2: Photographs of a pig lung. Figure S3: Comparison in the brightness of ICG in different media. Figure S4: Molecular structure and absorption peak of IR-780. Figure S5: The scheme of lung airways, generations, and bronchus diameters. Table S1: The fluorescence imaging depths in lung tissues with ICG gels of different concentrations as the imaging contrast. References [43,44] are cited in the supplementary materials.

**Author Contributions:** Z.W. (Zongyu Wu): Methodology, Formal analysis, Investigation, Data curation, Writing—original draft, Visualization. Z.W. (Ziwen Wang): Investigation, Formal analysis, Writing—original draft, Visualization. H.X.: Software, Formal analysis, Investigation, Data curation. Y.W.: Investigation, Data curation, Visualization. H.H.: Methodology, Visualization. S.N.: Conceptualization, Resources, Supervision. Writing—review and editing. J.Y.: Supervision, Validation, Resources, Funding Acquisition, Writing—review and editing. L.L.: Conceptualization, Project administration, Visualization, Validation, Funding acquisition, Writing—review and editing. Z.W. (Zongyu Wu) and Z.W. (Ziwen Wang) contributed equally. All authors have read and agreed to the published version of the manuscript.

**Funding:** This work was supported by National Natural Science Foundation of China (Nos. 81871401 and 81901786), the Science and Technology Commission of Shanghai Municipality (No. 19441905300 and 21511102100), Shanghai Jiao Tong University (Nos. YG2019QNA28 and YG2022QN006) and Shanghai Key Laboratory of Gynecologic Oncology.

**Institutional Review Board Statement:** Not applicable.

**Informed Consent Statement:** Not applicable.

**Data Availability Statement:** Not applicable.

**Conflicts of Interest:** The authors declare no conflict of interest.

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
