# Peer review of "Raman-Guided Bronchoscopy: Feasibility and Detection Depth Studies Using Ex Vivo Lung Tissues and SERS Nanoparticle Tags"

_photonics, doi:10.3390/photonics9060429_

Round 1

Reviewer 1 Report

In "Raman-Guided Bronchoscopy: Feasibility and Detection Depth Studies Using Ex-Vivo Lung Tissues and SERS Nanoparticle Tags," the authors produce SERS NPs and disperse them in tissue phantoms dubbed "SERS gels" that are embedded into pig lung tissue. These SERS gels are then probed by a home-built fiber-optic probe system, and the signal detection depth is characterized. The system is then compared with fluorescence measurements using ICG. Overall this is a relatively routine report, but may be of interest to others working on SERS NPs and surgical navigation or endoscopy. However, some issues must be addressed before publication could be considered:

1. The authors compare the SERS NPs performance to similar phantoms with ICG. However, this comparison is relatively meaningless due to the wide variation between the detection setups. The utilized power densities between the Raman and fluorescence systems are different by more than 4 orders of magnitude, and the exposure times are different by nearly an order of magnitude, leading to a ~100,000-fold difference between the two systems. That also neglects the difference in the detection systems, as the Raman detector is almost surely deeply cooled and less noisy/more sensitive than the camera used on the commercial surgical navigation system. Why this was done is especially confusing because as an NIR dye, ICG should be able to be easily excited by the Raman system utilized in this study. I suggest the authors to re-measure the ICG data using a similar method to the SERS gel method. Create ICG-gel beads and insert them into the pig tissue and then detect using the same fiber optic probe. This would seem to be a much more straightforward and meaningful comparison between the two reporters. This is especially important because ICG has the advantage of easy delivery to the lesion, and FDA approval. SERS NPs, as the authors state, are generally not biocompatible, and once encapsulated in silica shells are extremely difficult to easily and accurately deliver to tumors.

2. The method of inserting a SERS gel, then removing it and testing the next distance seems to be very inaccurate. Once a SERS gel has been inserted (say at the 1mm position), and then removed, and a new bead inserted at the 2mm position, the tissue at the 1mm position has been irrevocably altered. There are now air or fluid gaps and the tissue structure where the gel bead was inserted has been destroyed, potentially biasing the results. It would be better to make, say, 10 measurements at each bead distance using different locations on the lung for each measurement to avoid this issue.

3. The authors report a certain penetration depth in tissue, which has relevance only for their specific SERS tag, and their specific detection system. Other systems with higher or lower SERS intensities, and higher or lower excitation powers and detection efficiencies, would surely achieve different sensitivities. Therefore, the major contribution of this paper is not on the calculation or determination of specific penetration depths (the current focus of the article), but actually on the influence of the wall and the decay of SNR vs. distance. These are properties of the lung tissue and would be more generally true no matter which detection system or probe was used. In other words, if the SNR was 6 at an initial probe depth for a given probe and detector system, then the authors' slope calculation would allow a researcher to estimate their system's ultimate sensitivity vs. depth. I suggest the article be rephrased to emphasize this point.

4. There are minor language issues throughout the text but these mostly don't interfere with the scientific accuracy. However, the authors should try and revise the manuscript carefully before resubmission.

To conclude, the authors present several experiments aimed at determining how SERS signals decay versus tissue depth within lung tissues for the purposes of future endoscopic or surgical guidance. These are compared with fluorescence-based signals using the FDA-approved contrast agent ICG. While the results may have some limited relevance to others working in the same field, there are serious issues with the experimental design that must be addressed via new experiments prior to the paper being acceptable for publication.

Reviewer 2 Report

This work demonstrates SERS detection of Lung tissues.  The reviewer wonders if this technique can really be applicable to lung cancer detection or not.  However, the experiment on ex-vivo tissue may be successful.  This work may be acceptable after revisions as follows;

1.    The uniformity of distribution of nanoparticles should be secured.  It is because the objective of this work aims to detect lung cancer which occurs in sparsely in the lung.  So, if the distribution of nanoparticle is not smoothly and uniformly distributed in the tissue, the operator may miss the cancer.  How the authors can say the uniformity of detection?

2.    The spectral pattern of SERS strongly depends on the spectral measurements.  Is the reproducibility of SERS spectra is secured for different lots of nanoparticles?

Reviewer 3 Report

This work reports on the ex-vivo detection of phantom tumor in fresh pig lung tissues by using a miniaturized Raman device guided bronchoscopy. Results show that the SERS-guided bronchoscopy can achieve a detection through up to 5-mm lung tissues and 0.85-mm bronchi wall, due to the high sensitivity of SERS-active nanoparticles. Together with fluorescence imaging, the combinatory techniques allow not only ultra-small micro surgical sites on patients, but also straightforward wide-field visualization of tumor regions. This work demonstrates the feasibility of SERS spectroscopy in minimally invasive imaging and detection of lung lesions. I am more than pleased to see the advance of SERS spectroscopy in ex-vivo detection. This work should be published in the journal of Photonics.

Reviewer 4 Report

Manuscript Photonics #1749167

General comments

Quality of Scientific/Technical Content

The authors report about the use of a miniaturized Raman device that is inserted into one of the bronchoscope channels for sensitive detection of “phantom” tumors using fresh pig lung tissues and surface-enhanced Raman scattering (SERS) nanoparticle tags.

The proposed approach is original and the proposed method, if assessed, would give a contribution in the field.

The proposed experiments would be appropriate to proof the method, but the experimental details should be better clarified.

The work should be better placed in the scientific context especially considering that the general audience of the journal.

Data analysis should be improved.

Quality of Presentation

The title is accurate and identify the subject matter.

The abstract is fully comprehensible.

There are some typing errors.

Some specific comments

- Line 79. The usefulness of ex vivo studies should be better discussed.

- Line 129. How the authors evaluated the bead dimensions?

- Line 153. The description of the experimental procedures would benefit of the use of Fig.2. It should be introduced here.

-Eq.1. Some references should be added to explain why the authors select this value (3) as limit for the ratio

-Lines 159-161. The reasons for the selection of the listed peaks (525, 559 and 1206 cm-1) and for the spectral range for evaluating the noise should be given here.

-Line 177: Please explain why the reported integration time and gain were used.

-Lines 203-206: these details are redundant

-Lines 212-215. Are these results referred to measurements of the spectra directly form the beads? Please better explain how the measurements were performed

-Line 221 and Figure S5: It is better to represent spectra in Figure by using averages and shadows for standard deviations. In the present form it is difficult to appreciate the claimed stability

Lines 258-260: Please add a piece of discussion about the effects of the tumour absorption.

Lines 297-303 and Fig.3. Discussion about the experimental errors on the experimental data points used for the fitting procedures should be added. The description of the data analysis results is poor. An exponential fitting and a linear fitting procedure are performed on a very low number of points. It should be discussed. The reported values of R2 have to be discussed also considering the in one case the fit was performed using only 4 points. The obtained values of intercepts and slopes should be given adding the errors. The discussion should be rewritten considering the parameter errors.

-Fig.3 and discussion. The evidence that two over three of the selected peaks are not useful because of the presence of optical fibre signal should be better discussed otherwise a reader would think that the optical material was not well selected.

-Line 348: Why the reference signal is called “the noise of the background”?

-Lines 371-373. The statement is misleading. The possibility of using a 5s-acquisition time for SERS is not really an advantage put it is a need, otherwise the signal would be too low. Please better explain also the claimed higher stability of the SERS as compared to organic fluorophores.

Lines 138-139 and 397-402. The question of the very high power-density should be better discussed. Presently, it is a very significant element limiting the importance of the results.

Round 2

Reviewer 1 Report

The authors present a revised version of their manuscript describing the detection limit of SERS NPs in lung tissues using gel-bead phantoms. While many points of the manuscript are greatly clarified, particularly with respect to the experimental measurements (which confused both reviewers in different points), the authors did not adequately address the SERS vs. ICG sensitivity study. The authors make some vague arguments in their response letter regarding photobleaching and autofluorescence background being a reason why the ICG could not be compared using the SERS detection device. However, these arguments do not hold up to scrutiny. If the photobleaching was so serious, why not simply adjust the laser intensity down slightly? And given that the commercial system was able to visualize ICG fluorescence without autofluorescence interference, it follows that your system should be able to do the same.

To get around this issue, the authors amend the manuscript to clarify that the SERS vs. ICG is not a direct comparison, but if it's not a direct comparison, what value does it add to the study? The depth limit claimed by the authors for ICG is not comparable to the depth limit claimed for SERS, so why bother reporting it? I think the authors share my intuition that in an apples-to-apples comparison, ICG would be more sensitive than SERS. SERS and Raman techniques more generally have many wonderful properties to promote their usage, but signal strength compared to fluorescence is certainly not one of them.

I recommend the authors either fully remove this section if they are not willing to perform a true apples-to-apples comparison, or else they should continue to amend the manuscript to clarify that the achieved depth sensitivity via ICG is valid only for the utilized measurement device. And, actually, if a more sensitive fiber probe system was utilized, the measurement device would likely be higher, and potentially significantly higher than SERS. An example would be a not-so-recent paper from the Annals of Thoracic Surgery (10.1016/j.athoracsur.2014.07.050) where ICG was detected to 12mm in a deflated lung (equivalent to ~24mm in an inflated lung) using a "bead insertion" phantom very similar to the one utilized here. Therefore, the authors highlighting their technique's supposed advantage over ICG both in the abstract and in the text is highly inappropriate.

Given this serious deficiency in the experiments and text, I cannot recommend publication at this time.

Reviewer 4 Report

Despite some of the raised points have been adequately faced, some others remain open. In particular, the questions of the low number of points used for fitting procedures, lack of information about the errors on the obtained parameters and the not conclusive discussion about the values of R2 have not been properly considered. The statement "the two lines share similar slopes (-sigma), as -0.83 and -0.89, respectively" well represents the incompleteness of the discussion. What is the meaning of "similar values"? What about errors? What about methods for comparing experimental values (including errors)?

These aspects have further gained importance considering that the without-error slopes are used for further estimations discussed in detail (see lines 346-361).

The answer to the point about Line 221 and Figure S5: It is better to represent spectra in Figure by using averages and shadows for standard deviations. In the present form it is difficult to appreciate the claimed stability” is not satisfying.

The question of the very high power-density still deserves to be better discussed.

There are some new typing errors.

The numbering of the references should be checked (for instance, reference 28 seems wrongly cited).
